# Knockdown of Vacuolar ATPase Subunit G Gene Affects Larval Survival and Impaired Pupation and Adult Emergence in *Henosepilachna vigintioctopunctata*

**DOI:** 10.3390/insects12100935

**Published:** 2021-10-14

**Authors:** Jie Zeng, Wei-Nan Kang, Lin Jin, Ahmad Ali Anjum, Guo-Qing Li

**Affiliations:** Agriculture Ministry Key Laboratory of Integrated Pest Management on Crops in East China/State & Local Joint Engineering Research Center of Green Pesticide Invention and Application, Department of Entomology, College of Plant Protection, Nanjing Agricultural University, Nanjing 210095, China; 2018102081@njau.edu.cn (J.Z.); 2017202041@njau.edu.cn (W.-N.K.); jinlin@njau.edu.cn (L.J.); 2018202068@njau.edu.cn (A.A.A.)

**Keywords:** *Henosepilachna vigintioctopunctata*, vacuolar ATPase subunit G, larval survival, gut, Malpighian tubule

## Abstract

**Simple Summary:**

Vacuolar ATPase (vATPase), a proton pump driven by ATP hydrolysis, acts as a membrane energizer to motivate the movement of ions and nutrients across the cellular membrane in insect guts and Malpighian tubules, among others. The vATPase holoenzyme contains 16 subunits. Out of these subunits, mammalian G subunit includes three isoforms (G1-G3) which are encoded by three distinctive genes. The physiological role of a specific G isoform can be compensated by others. Thus, current experimental evidence on the in vivo function of G is rather limited among eight V_1_ subunits. In the present paper, particular attention was paid to an insect model, *Henosepilachna vigintioctopunctata* ladybird, a serious defoliator of Solanaceae and Cucurbitaceae plants in many Asian countries. Given that the beetle is sensitive to RNA interference (RNAi), *HvvATPaseG* gene was knocked down by ingestion of its corresponding dsRNA at the fourth-instar larval stage. Silence of *HvvATPaseG* affected larval growth and survival, impaired pupation and adult emergence. Our results provide a basis for further functional research on the vATPase G subunit in insects and suggest new ideas for the management of *H. vigintioctopunctata*.

**Abstract:**

The vATPase holoenzyme consists of two functional subcomplexes, the cytoplasmic (peripheral) V_1_ and the membrane-embedded V_0_. Both V_1_ and V_0_ sectors contain eight subunits, with stoichiometry of A_3_B_3_CDE_3_FG_3_H in V_1_ and ac_8_c’c”def(Voa1p) in V_0_ respectively. However, the function of G subunit has not been characterized in any non-Drosophilid insect species. In the present paper, we uncovered that *HvvATPaseG* was actively transcribed from embryo to adult in a Coleopteran pest *Henosepilachna vigintioctopunctata*. Its mRNA levels peaked in larval hindgut and Malpighian tubules. RNA interference (RNAi)-mediated knockdown of *HvvATPaseG* significantly reduced larval feeding, affected chitin biosynthesis, destroyed midgut integrity, damaged midgut peritrophic membrane, and retarded larval growth. The function of Malpighian tubules was damaged, the contents of glucose, trehalose, lipid, total soluble amino acids and protein were lowered and the fat bodies were lessened in the *HvvATPaseG* RNAi larvae, compared with those in the PBS- and ds*egfp*-fed beetles. In contrast, the amount of glycogen was dramatically increased in the *HvvATPaseG* depletion ladybirds. As a result, the development was arrested, pupation was inhibited and adult emergence was impaired in the *HvvATPaseG* hypomorphs. Our results demonstrated that G subunit plays a critical role during larval development in *H. vigintioctopunctata*.

## 1. Introduction

Vacuolar-type H^+^-ATPase (vATPase) is a proton translocating pump present in the internal membranes of all eukaryotic cells [1,2]. By hydrolyzing ATP to ADP and phosphate, vATPases transport protons across the cell membrane to generate a membrane potential (V_m_) as high as 120 mV [3,4]. The V_m_ can motivate transmembrane movement of Cl^−^ through a Cl^−^ channel and maintain pH gradients in specific organelles such as Golgi, endosomes, lysosomes or secretory vesicles in eukaryotic cells [5,6]. Through maintenance of pH gradients in specific cellular organelles, vATPases exert critical roles in numerous biological processes, such as endocytosis, autophagy, protein processing and degradation, amino acid transportation, synaptic vesicles loading and coupled transport, and waste disposal [7,8,9,10]. Accordingly, loss of vATPase activity results in embryonic lethality in mammals [11].

Moreover, vATPase also contributes to establish V_m_ outside the differentiated cells, for instance insect epithelial tissues [12,13]. The transmembranous potential V_m_ then activates secondary K^+^/2H^+^-antiporter to secret K^+^ into the lumens of the guts, Malpighian tubules, ovarioles, testes, salivary glands, labial glands and sensory sensilla [14,15]. In insect gut lumens, a high level of K^+^ plays two important physiological roles. Firstly, the high K^+^ content facilitates secretion of HCO_3_^−^ across the apical membrane; HCO_3_^−^ is subsequently transferred to CO_3_^−^. CO_3_^−^ provides the main anions to alkalinize the luminal contents to pH above 10 [16,17,18], an appropriate alkaline condition to maintain the activities of numerous digestive enzymes in the insect midguts [19]. Secondly, the transmembranous K^+^ gradient of the midgut epithelial cells activates K^+^-dependent uptake of amino acids via a nutrient amino acid transporter [3,14]. In insect Malpighian tubules, a high level of K^+^ in the lumen drives a passive movement of anions, such as Cl^−^, HCO_3_^−^ and perhaps urate ions to form primary urine [6,14].

The structures of vATPases are conserved in eukaryote species [10,20]. The vATPase holoenzyme is a multi-subunit complex that comprises two functional sectors, V_1_ and V_0_. The cytoplasmic (peripheral) V_1_ subcomplex is involved in ATP hydrolysis and consists of eight subunits (A-H), with stoichiometry of A_3_B_3_CDE_3_FG_3_H in Lepidopteran *Manduca sexta* and yeast *Saccharomyces cerevisiae*. V_0_ serves as a proton pump driving transmembrane proton movement. It contains eight subunits, with stoichiometry of ac_8_c’c”def(Voa1p) [1,20,21,22,23,24]. In the V_1_ sector, an (AB)_3_ hexamer forms an ATPase motor. The (AB)_3_ hexamer is connected to the V_0_ proton-transporting sector by a central stalk composed of the D subunit and three elongated structures. Each elongated structure is made up of an E and a G subunit [1,2].

However, current experimental evidence on the in vivo function of G is rather limited among 8 V_1_ subunits [22,25]. Mammalian G subunit includes three isoforms (G1-G3) which are encoded by three distinctive genes namely *ATP6V1G1*, *ATP6V1G2* and *ATP6V1G318* [26,27]. These genes are expressed in a tissue- and cellular-compartment-dependent way [28,29]. However, the physiological role of a specific G isoform can be compensated by others. In the *G2*-null mouse brain, for example, the *G1* isoform abundantly accumulates to complement the function of neuron-specific *G2* isoform. As a result, the *G2* null mutant shows no apparent disorders in neuron architecture and behavior [4].

It appears that mammals are not the suitable model animals to explore the physiological roles of G subunit. Therefore, a particular attention should be paid to the insect models. In *M. sexta*, G subunit has been reported as a peripheral vATPase subunit [30,31]; it stimulates the ATPase activity of the reassembled V_1_ complex [32]. In *D. melanogaster*, only one gene (*vha13*, CG6213) encodes G subunit [33]. The expression of *vha13* is regulated by a basic helix-loop-helix leucine zipper transcription factor Mitf through an M-box *cis*-site [(T/G)GTC(A/T)TGTGA] [34]. Moreover, mutation in *vha13* causes embryo lethality and a transparent Malpighian tubule phenotype [33]. However, the function of G subunit has not been characterized in any non-Drosophilid insect species.

*Henosepilachna vigintioctopunctata* is a serious defoliator of Solanaceae and Cucurbitaceae plants in many Asian countries [35]. The beetle is sensitive to RNA interference (RNAi) [36,37,38,39]. This offers a great opportunity to explore the molecular mechanisms of *vATPaseG* gene by RNAi. In the present paper, we discovered that knockdown of *HvvATPaseG* affected larval growth and survival, and impaired pupation and adult emergence. It appears that the G subunit exerts a critical physiological role during larval development in *H. vigintioctopunctata*.

## 2. Materials and Methods

### 2.1. Insect

*H. vigintioctopunctata* adults were collected from *Solanum melongena* L. in Nanjing city, Jiangsu Province, China, in the summer of 2018. The beetles were routinely maintained in an insectary at 28 ± 1 °C under a 16 h:8 h light-dark photoperiod and 50–60% relative humidity using foliage at the vegetative growth or young tuber stages in order to assure sufficient nutrition. At this feeding protocol, the larvae progressed through four distinct instars, with approximate periods of the first-, second-, third-, and fourth-instar stages of 3, 2, 2 and 3 days, respectively. Upon reaching full size, the fourth larval instars stopped feeding, fixed their abdomen ends to the substrate surface and entered the prepupal stage. The prepupae spent an approximately 2 days to pupate. The pupae lasted about 4 days and the adults emerged.

### 2.2. Molecular Cloning

To identify *vATPaseG* gene, a TBLASTN search of the transcriptome data [35] was performed using the amino acid sequences of *D. melanogaster* vATPaseG (XP_022216364.1)) as query. This resulted in the identification of a putative *HvvATPaseG*.

TRIzol reagent (Invitrogen, New York, NY, USA) was used to extract the total RNA in accordance with the manufacturer’s protocols. The NanoDrop 2000 spectrophotometer (Thermo Fisher Scientific, New York, NY, USA) was used to perform the RNA quantification. RNA purity was determined by assessing optical density (OD) absorbance ratios at OD260/280 and OD260/230. The integrity of RNA was analyzed via 1% agarose gel electrophoresis with ethidium bromide staining. Reverse transcription was performed using a PrimeScript^TM^ RT reagent Kit with a gDNA Eraser (TaKaRa Biotechnology Co. Ltd., Dalian, China). Briefly, reaction was incubated at 37 °C for 15 min and then 85 °C for 5 s. The resultant cDNA was preserved at −20 °C for further use.

The correctness of the sequences was substantiated by polymerase chain reaction (PCR) and sequencing using primers in Appendix A. The sequenced cDNA was submitted to GenBank (accession number: *vATPaseG*, MW267253). The protein sequences of vATPaseG from three Dipteran *Lucilia cuprina*, *Drosophila melanogaster* and *Aedes aegypti*, two Coleopteran *Anoplophora glabripennis* and *Agrilus planipennis*, two Lepidopteran *Helicoverpa armigera* and *Spodoptera litura*, two Hymenopteran *Bombus impatiens* and *Harpegnathos saltator*, two Hemipteran *Halyomorpha halys* and *Acyrthopiphon pisum*, and a Thysanopteran *Frankliniella occidentalis* were acquired from GenBank (http://www.ncbi.nlm.nih.gov/, accessed on 21 April 2021). Phylogenetic analysis was conducted using MEGA-5 software and the neighbor-joining method with 1000 bootstrap replications.

### 2.3. Preparation of DsRNAs

A cDNA fragment derived from *Hv**vATPaseG* was selected, and was further BLAST (BLASTN) searched against the *H. vigintioctopunctata* transcriptome data [35] to identify any possible off-target sequences that had an identical match of 20 bp or more. A cDNA from the enhanced green fluorescent protein (*egfp*) gene from *Aequorea victoria* was used as control. Specific primers used to clone the fragments of dsRNAs were listed in Appendix A. These dsRNAs were individually expressed using *Escherichia coli* HT115 (DE3) competent cells lacking RNase III following the established method [38]. Individual colonies were inoculated and grown until cultures reached an OD600 value of 1.0. The colonies were then induced to express dsRNA by addition of isopropyl β-D-1-thiogalactopyranoside to a final concentration of 0.1 mM. The expressed dsRNA was extracted and confirmed by electrophoresis on 1% agarose gel (Appendix A). Bacteria cells were centrifuged at 5000× *g* for 10 min, and resuspended in an equal original culture volume of 0.05 M phosphate buffered saline (PBS, pH 7.4). The dsRNA concentration recovered from agarose gel was determined using an ultramicrospectrophotometer (ThermoFisherNanoDrop One, the Georgia World Congress Center, Atlanta, GA, USA) and the dsRNA content in the bacterial solution was estimated at about 0.5 μg/mL. The bacterial solutions were used for experiment.

### 2.4. Dietary Introduction of DsRNA

A similar method, as previously reported, was used to introduce dsRNA into the *H. vigintioctopunctata* larvae [40]. Potato leaves were immersed with a bacterial suspension containing ds*vATPaseG* for 5 s, removed, and dried for 2 h under airflow on filter paper. The PBS- and ds*egfp*-dipped leaves were used as controls. Five treated leaves were then placed in Petri dishes (9 cm diameter and 1.5 cm height). The newly-ecdysed fourth-instar larvae were starved for at least 4 h prior to the experiment. Then, ten larvae were transferred to each dish as a repeat. For each treatment, 12 repeats were set. Three replicates were used to observe the survival, pupation and emergence by allowing the larvae to feed on treated leaves for 3 days (replaced with freshly treated ones each day), and on untreated foliage until reaching the prepupal stage. Three replicates were continuously fed on treated foliage for 3 days and were collected for extraction of total RNA. The other six replicates were used for gut dissection, staining and determination of protein, total amino acids, triglyceride and sugars.

### 2.5. Real-Time Quantitative PCR (qRT-PCR)

For temporal expression analysis, RNA templates were derived from eggs (day 3), the larvae from the first through the fourth instars, prepupae, pupae and adults. For analysis of the tissue expression patterns, RNA templates were from the foregut, midgut, hindgut, Malpighian tubules, epidermis and fat body of the day 1 final instar larvae. Each sample contained 20–30 individuals and repeated three times. For analysis of the effects of treatments, total RNA was extracted from treated larvae. Each sample contained 10 individuals and repeated three times. The RNA was extracted using an SV Total RNA Isolation System Kit (Promega). Purified RNA was subjected to DNase I (Roche, Basel, Switzerland) to remove any residual DNA according to the manufacturer’s instructions. Quantitative mRNA measurements were performed by qRT-PCR in technical triplicate, using 2 internal control genes (*HvRPS18* and *HvRPL13*, the primers listed in Appendix A) according to the published results [41]. An RT negative control (without reverse transcriptase) and a non-template negative control were included for each primer set to confirm the absence of genomic DNA and to check for primer-dimer or contamination in the reactions, respectively.

According to a previously described method [42], the generation of specific PCR products was confirmed by gel electrophoresis. The primer pair for each gene was tested with a five-fold logarithmic dilution of a cDNA mixture to generate a linear standard curve (crossing point [CP] plotted vs. log of template concentration), which was used to calculate the primer pair efficiency. All primer pairs amplified a single PCR product with the expected sizes, showed a slope less than −3.0, and exhibited efficiency values ranging from 2.4 to 2.7. Data were analyzed by the 2^−ΔΔCT^ method, using the geometric mean of the four internal control genes for normalization.

### 2.6. Hematoxylin-Eosin (HE) Staining

HE staining was performed to observe defective phenotypes in guts, Malpighian tubules and head capsule. Briefly, the three portions in the PBS-, ds*egfp*- and ds*vATPaseG*-fed larvae were dissected 4 days after the initiation of bioassay, and were then fixed in 4% paraformaldehyde and embedded in paraffin. The three embedded tissues were subsequently cut into 6μm-thick sections. The sections were stained using Mayer’s H&E (Yeasen, Shanghai, China) following a routine staining procedure and observed with an Olympus BH-2 light microscope (Olympus, Tokyo, Japan).

### 2.7. TUNEL Assay

To detect apoptotic cells, midgut sections were examined for DNA fragmentation with the TUNEL in situ cell death detection kit (Roche Applied Science, ON, Canada) following the manufacturer’s instructions. DAPI was used to counter-stain nuclei. Apoptotic cells were imaged under a fluorescence microscope (Olympus BX51; Leica TCS SP8 STED) under differential interference contrast using the appropriate filter.

### 2.8. EdU Labeling

The proliferation cells were detected by the Click-iT EdU Imaging Kit (Invitrogen, Carlsbad, CA, USA). The midguts were fixed with 4% paraformaldehyde, rinsed twice with 3% BSA in PBS and permeabilized with a 0.5% Triton^®^ X-100 in PBS. The Click-iT^®^ reaction cocktail was freshly prepared according to the manufacturer protocol. Briefly, 430 μL of 1× Click-iT reaction buffer was mixed with 20 μL of CuSO4 solution, 1.2 μL Alexa Fluor^®^ azide and 50 μL click reaction additive (sodium ascorbate) to reach a final volume of about 500 μL per coverslip. Next, the midgut sections were stained by incubating in the dark for 30 min with Click-iT^®^ reaction cocktail at room temperature. After staining, the midgut sections on coverslips were washed several times with 3% BSA in PBS. For nuclear staining, the Hoechst 33342 (Component G) solution was diluted at the ratio of 1:2000 in PBS to obtain a 1X Hoechst 33342 solution (the final concentration was 5 µg/mL), the solution was used to counter-stain nuclei. The images were taken under a fluorescence microscope (Olympus BX51; Leica TCS SP8 STED) under differential interference contrast using the appropriate filter.

### 2.9. Determination of Nutrients

Protein concentrations were determined by the Bradford assay [43], using a total protein quantitative kit (Nanjing Jiancheng Science and Technology Co., Ltd., Nanjing, China) and bovine serum albumin (BSA) as standards. Briefly, the whole bodies of the collected samples were ground with 3 mL pH 7.8 PBS in an ice bath and then centrifuged at 4 °C, 7400× *g* for 10 min (Eppendorf centrifuge 5417R, Hamburg, Germany). The supernatants were used to analyze the protein following the Coomassie brilliant blue G-250 method using a spectrophotometer (Hitachi U-3310, Tokyo, Japan). Five 0.1–1.0 mg/mL BSA protein standards were prepared. Both protein standards and samples were added to 5 mL Coomassie brilliant blue G-250 dye reagent, incubated at room temperature for 5 min, and then the absorbance was measured at 595 nm.

The levels of total amino acids and triglyceride were measured by commercially available kits (Nanjing Jiancheng Science and Technology Co., Ltd., Nanjing, China), according to the manufacturer’s instruction.

Glycogen, trehalose and glucose contents were measured as described previously [44]. The samples were homogenized in 250 mL 0.25 M sodium carbonate buffer. The mixture was incubated at 70 °C for 10 min, and then 150 mL 1 M acetic acid and 600 mL 0.25 M sodium-acetate (pH 5.2) were added. For each sample, four portions with an identical weight were prepared. The first two portions were mixed with or without 1 unit α-amyloglucosidase (Sigma), and incubated for 4 h at 40 °C. The second two portions were incubated overnight at 37 °C with or without 1 μL porcine kidney trehalase (Sigma, Darmstadt, Germany). Glucose contents were quantified according to the manufacturer’s instructions (Glucose Assay (GO) Kit; Sigma, Darmstadt, Germany). Free glucose calculated from the samples without amyloglucosidase or trehalase treatment was subtracted from the samples with amyloglucosidase or trehalase treatment to obtain the glycogen or trehalose value, respectively.

### 2.10. Data Analysis

We used SPSS for Windows (Chicago, IL, USA) for statistical analyses. The averages (±SE) were submitted to analysis of variance with the Tukey-Kramer test. Moreover, no significant differences between dsRNAs targeting two different regions of *Hv**vATPaseG* gene (ds*F*-1, ds*F*-2) were present; the data of each gene were thus combined.

## 3. Results

### 3.1. Identification of HvvATPaseG

An *HvvATPaseG* transcript was identified from the transcriptome data [35]. Polymerase chain reaction (PCR) and sequencing was performed and the correctness of *HvvATPaseG* mRNA sequence was confirmed.

Thirteen vATPase subunit G proteins were used to construct an unrooted phylogenetic tree using the neighbor-joining method. These sequences were from three Dipteran *Lucilia cuprina*, *Drosophila melanogaster* and *Aedes aegypti*, three Coleopteran *Henosepilachna vigintioctopunctata*, *Anoplophora glabripennis* and *Agrilus planipennis*, two Lepidopteran *Helicoverpa armigera* and *Spodoptera litura*, two Hymenopteran *Bombus impatiens* and *Harpegnathos saltator*, two Hemipteran *Halyomorpha halys* and *Acyrthopiphon pisum*, and one Thysanopteran *Frankliniella occidentalis*. The tree revealed that vATPaseG proteins formed Hemipteran, Hymenopteran, Coleopteran, Dipteran, Lepidopteran and Thysanopteran clades. However, the placement of the protein from *F. occidentalis* was uncertain due to low bootstrap values. In the Coleopteran subclade, *Hv*vATPaseG first clustered with that from *A. planipennis*; and was then joined together with that from *A. glabripennis*, with 86% bootstrap support (Appendix A).

### 3.2. Expression Profiles of HvvATPaseG

In order to determine the temporal expression profiles, qRT-PCR was performed. *HvvATPaseG* was widely expressed throughout developing stages, from embryo (egg) to adult. It was peaked and troughed at the day 1 fourth-instar stage and the pupal period, respectively. Moreover, *HvvATPaseG* was abundantly transcribed at the early and/or late stage of the first-, second-, third- and fourth-instar larvae (Figure 1A).

To test at which tissues *HvvATPaseG* is expressed and potentially required, the mRNA levels of *HvvATPaseG* were measured in the foregut, midgut, hindgut, Malpighian tubules, epidermis and fat body of the day 1 fourth-instar larvae. The mRNA of *HvvATPaseG* was detectable in all the tested tissues. The transcript levels were high in the hindgut and Malpighian tubules, intermediate in the foregut and midgut and low in the epidermis and fat body (Figure 1B).

### 3.3. Ingestion of DsvATPaseG Affects Larval Performance

We dietarily introduced ds*vATPaseG* into the newly-molted fourth-instar larvae in order to explore the physiological roles. After three days’ ingestion of ds*vATPaseG*, the mRNA level of *HvvATPaseG* was significantly downregulated (Figure 2A). Moreover, larval growth was inhibited when measured by the larval fresh weight, compared with the PBS- and ds*egfp*-fed beetles (Figure 2B. Consequently, the larval sizes of the *HvvATPaseG* hypomorphs were smaller than those in the ds*egfp*-fed ones (Figure 2F,G: ds*egfp*-fed beetle (left) v.s. treated one (right)).

The PBS- and ds*egfp*-ingested beetles routinely pupated and emerged as adults, with little larval mortalities (Figure 2C–E,H). In contrast, approximately 90% of the ds*vATPaseG*-fed larvae failed to pupate (Figure 2C), remained as stunting prepupae (Figure 2I–N) and died within ten days.

Less than 10% of the *HvvATPaseG* depleted larvae normally pupated and around 20% of the resultant pupae emerged as adults (Figure 2D,E). The *HvvATPaseG* depleted adults looked normal, but they eventually died within 1 week after emergence.

### 3.4. The Integrity of the Midgut after RNAi

To examine the integrity of the midgut, the treated larvae after consumption of PBS, ds*egfp* or ds*vATPaseG* for 3 days, and normal foliage for an additional of one day were dissected (Figure 3). The guts from the PBS- and ds*egfp*-fed beetles were full of food. In contrast, the guts from the ds*vATPaseG*-fed larvae were almost empty. The average lengths of the guts of the PBS-, ds*egfp*- and ds*vATPaseG*-fed beetles were 16.6, 17.3 and 8.2 mm; the average midgut widths were 0.55, 0.53 and 0.31 mm. The former two were significantly different than the later (Figure 3A and Appendix A).

Figure 2B–H showed the results of HE staining of dissected guts 1–4 days after the initiation of bioassay. In the PBS- and ds*egfp*-fed groups, the principal cells of the midgut were dense, tall and columnar; the peritrophic membrane (PM) formed a lining layer that separated the food from the midgut epithelium in the actively feeding larvae (Figure 3B). At the late stage of the fourth instar when the larvae stopped feeding, the food circling by PM became less (Figure 3C). After gut clearing just before ecdysis, the midgut lumens were empty (Figure 3D).

Ingestion of ds*vATPaseG* damaged the integrity of the midgut. Three to four days after dsRNA ingestion, the principal cells of the midgut were sparse, with abnormal forms. The PM in the midgut disappeared. Many dying cells separated from the midgut epithelium and moved to the lumens (Figure 3E–G). Meanwhile, TUNEL positive cells were detected in the midguts dissected from the insects having ingested ds*vATPaseG*, in contrast to midgut cells from PBS- and ds*egfp*-fed groups (Figure 3J vs. Figure 3I).

The red EdU-positive cells implied actively dividing cells in the midgut. These cells were present in the midguts in the ds*vATPaseG*-fed larvae, in contrast to the PBS- and ds*egfp*-fed beetles (Figure 3L vs. Figure 3K).

### 3.5. RNAi of HvvATPaseG on Malpighian Tubules

To examine whether knockdown of *vATPaseG* causes a transparent Malpighian tubule phenotype, as that seen in *D. melanogaster* [45], the treated larvae after consumption of ds*vATPaseG* for 3 days and the normal foliage for an additional of 2 days were dissected and the Malpighian tubules were collected. The phenotypes were examined (Figure 4).

The Malpighian tubules from ds*egfp*-fed larvae were opaque. In contrast, the tubules in ds*vATPaseG*-fed larvae were clear (Figure 4A). The HE staining showed that the tubules from ds*egfp*-fed larvae were full of urine (Figure 4B), whereas most of the tubules in ds*vATPaseG*-fed larvae were empty (Figure 4C–E).

### 3.6. Silencing HvvATPaseG on the Sugar Contents

After ingestion of dsRNA for 3 days and the normal leaves for 2 additional days, the contents of glucose, trehalose and glycogen in the whole larvae were tested (Figure 5A–C).

When compared to the control specimens, the content of glucose was greatly reduced in whole bodies from the *HvvATPaseG* hypomorphs, with the mean content only about a half of the value for control insects treated with ds*egfp* (Figure 5A). Similarly, the amount of trehalose was significantly decreased in the *HvvATPaseG* hypomorphs 3, 4 and 5 days after experiment. The mean contents were only a tenth of the value for ds*egfp*-fed insects (Figure 5B). In contrast, the glycogen content in the *HvvATPaseG* RNAi sample was significantly increased, compared with that in the PBS- and ds*egfp*-fed beetles (Figure 5C).

In insects, for example, in *L. decemlineata*, trehalose-6-phosphate synthase (*TPS*) is a trehalose biosynthesis gene and trehalases (*TRE*s) are trehalose degradation genes [46]. In the present paper, we identified *HvTPS*, soluble trehalase genes *HvTRE1a* and *HvTRE1b*, and membrane-bound trehalase gene *HvTRE2*. We noted that the expression level of *HvTPS* was not changed, whereas the mRNA levels of *HvTRE1a*, *HvTRE1b* and *HvTRE2* were significantly decreased in the *HvvATPaseG* RNAi sample, compared with those in the PBS- and ds*egfp*-fed beetles (Figure 5D–G).

In *L. decemlineata*, uridine diphosphate *N*-acetylglucosamine pyrophosphorylases (UAPs) is associated with the biosynthesis of chitin [47]. In this study, we found the expression level of *HvUAP* was greatly reduced in the *HvvATPaseG* RNAi specimen, compared with those in the PBS- and ds*egfp*-fed larvae (Figure 5H). Consistently, the body cuticle was almost transparent in the ds*vATPaseG*-treated larvae, in contrast to ds*egfp*-ingested ones (Figure 5J vs. Figure 5I). This indicates that ds*vATPaseG*-treated larvae possess thin cuticle

Since chitin is an essential component of the epidermal cuticle in insects [47], we examined the depth of strongly sclerotized head capsule using HE staining (Figure 5I,J). We found that the head cuticle (9.5 and 8.9 μm) were thicker in the PBS- and ds*egfp*-fed larvae than that in the *HvvATPaseG* hypomorphs (5.4 μm) (Appendix A).

### 3.7. Shortage of Lipid and Protein in the HvvATPaseG RNAi Larvae

The larval specimens 4 and 5 days after experiment and the prepupal sample were collected and dissected. It was noted that the larvae having fed on PBS had more fat bodies than the *HvvATPaseG* hypomorphs (Figure 6A,B). No fat body was present in the *HvvATPaseG* RNAi prepupae (Figure 6C).

We further tested the contents of triglyceride, total soluble amino acids and proteins 5 days after experiment. As expected, the data in the *HvvATPaseG* hypomorphs were significantly lower than those in the PBS- and ds*egfp*-fed larvae (Figure 6D–F).

## 4. Discussion

Up to date, function characterization of vATPase subunit G has been documented in only one insect species, *D. melanogaster* [33]. In the present paper, we identified a subunit G gene in a non-Drosophilid insect species, *H. vigintioctopunctata*. We uncovered that *HvvATPaseG* transcripts were detectable from embryo (egg) to adult; its mRNA level peaked at day 1 fourth-instar larvae. Similarly, *vha13* is expressed in the embryo, larvae, pupae and adults in *D. melanogaster* [33]. According to the temporal expression patterns, we knocked down *HvvATPaseG* using RNAi to characterize its functions during larval development in *H. vigintioctopunctata*.

### 4.1. Knockdown of HvvATPaseG Causes Developmental Arrest

Our findings showed that knockdown of *HvvATPaseG* delayed larval growth, inhibited pupation and impaired adult emergence in *H. vigintioctopunctata*. Consistent with our data, mutation in vATPaseG-encoding gene *vha13* leads to a lethal phenotype at the late embryo and first larval instar stages in *D. melanogaster* [33]. The similar phenotypes between *D. melanogaster* [33] and *H. vigintioctopunctata* (this study) indicate that the physiological roles of vATPase subunit G during development are conserved in insects.

However, mammalian G subunit includes three isoforms (G1-G3) [26,27] and their physiological roles can be mutually complemented [4]. Therefore, we carefully examined the defects in the *HvvATPaseG* hypomorphs in *H. vigintioctopunctata*, in order to explore the physiological importance of vATPase subunit G.

### 4.2. HvvATPaseG Is Required for Midgut Integrity

Three pieces of experiment evidence found in this research imply that *Hv*vATPaseG is required for midgut integrity. Firstly, our data revealed that the *HvvATPaseG* levels were high in Malpighian tubules and guts. In agreement with our result, insect vATPases have been reported to be localized in the apical membranes of guts in other insect species [15]. In *D. melanogaster*, for example, Val13 is located in the Malpighian tubules, midgut, hindgut and rectum [33].

Secondly, our results showed that midgut growth was inhibited and midgut integrity was compromised, with sparse principal cells in the *HvvATPaseG* hypomorphs in *H. vigintioctopunctata*. Furthermore, we observed many apoptotic cells in the midgut and in its lumen by TUNEL assay. It appears that active apoptosis after RNAi of *HvvATPaseG* reduces the number of midgut principal cells and compromises midgut integrity.

It is well known that vATPases are ubiquitous proton pumps that maintain pH gradients in specific cellular organelles [5,6], to exert their important functions in numerous biological processes [7,8,9,10]. Accordingly, we hypothesized that RNAi of *HvvATPaseG* inhibits these essential biological processes in midgut principal cells, and result in massive apoptosis of midgut epithelium cells.

In the present paper, we found the red EdU-positive cells in the midguts in the ds*vATPaseG*-fed larvae, rather than in the PBS- and ds*egfp*-fed beetles, implying more actively dividing cells in the midgut of the *HvvATPaseG* hypomorphs. Consequently, we argued that knockdown of *vATPaseG* greatly triggered apoptosis of the midgut principal cells. In order to restore the basic function of the midgut, the intestinal stem cells actively divide. Even so, the integrity of the midgut was not restored.

Lastly, our results revealed that the peritrophic membrane (PM) in the midgut disappeared in the *HvvATPaseG* hypomorphs in *H. vigintioctopunctata*. Our further observation on the expression of a chitin biosynthesis gene *UAP* [47] and the cuticle depth of head capsule demonstrated that deficiency of chitin is response for the disappeared midgut PM. In agreement with our results, the Chinese mitten crab *Eriocheir sinensis* fails to form a new epicuticle after depletion of *vATPaseB* mRNA [48].

### 4.3. HvvATPaseG Is Essential for Availability of Nutrients

The damaged larval guts in the *HvvATPaseG* hypomorph undoubtedly limit digestion of food. Consistently, the guts from the ds*vATPaseG*-fed larvae were almost empty, in contrast to the guts from the PBS- and ds*egfp*-fed beetles which were full of food. As a result, larval growth was delayed. In agreement with our result, disrupting chitin biosynthesis to damage larval guts significantly reduces foliage consumption in *L. decemlineata*. The resulting larvae have lighter fresh weights and smaller body sizes [47,49].

In this study, we discovered that the contents of glucose, trehalose, lipid, total soluble amino acids and protein were greatly reduced in the *HvvATPaseG* hypomorphs. Moreover, less fat body was presented. It is known that high pH value is essential for maintenance of high enzymatic activities in the midguts [19]. Upon knockdown of the *HvvATPaseG*, the membrane potential (V_m_) is not established; K^+^ is not accumulated in the gut lumen; the pH value is lowered. As a result, the activities of digestive enzymes are reduced in the midgut lumen and available nutrients such as glucose, trehalose and lipid are deficient in the *HvvATPaseG* hypomorphs in *L. decemlineata*. Moreover, three serine (intestains, IntB4/D4/E2) and one cysteine (chymotrypsin-like, SPS1A) proteases have been reported as digestive proteases in *L. decemlineata* [50,51,52,53]. The influence of low pH value on the activities of these proteases in the *HvvATPaseG* RNAi beetle needs further research to clarify.

In addition, lack of amino acids and protein may also result from the dysfunction of vATPase in the *HvvATPaseG* hypomorphs. It is well known that vATPase-driven accumulation of K^+^ in the midgut lumen can drive the nutrient amino acid transporter to absorb amino acids [3]. In the *HvvATPaseG* hypomorphs in *H. vigintioctopunctata*, dysfunction of vATPase inhibits the uptake of amino acids. Consequently, the contents of total soluble amino acids and protein were greatly reduced.

Conversely, the glycogen content was significantly higher in the *HvvATPaseG* RNAi sample. In line with our result, dysfunction of vATPase affects lysosomal acidification, disrupts lysosomes to digest macro molecular from the endocytic and autophagic pathways [20]. This leads to defective processing and the abnormal storage of nutrients that are essential for cell survival [54]. Similar disruption of cellular lysosomes should occur in the *HvvATPaseG* RNAi beetles. As a result, glycogen is accumulated in *H. vigintioctopunctata*.

### 4.4. HvvATPaseG Is Involved in the Formation of Urine in Malpighian Tubules

In this research, we reach a conclusion that *Hv*vATPaseG is associated with the formation of urine in Malpighian tubules based on three pieces of experiment evidence. Firstly, *HvvATPaseG* was highly expressed in the Malpighian tubules. Similarly, vATPases have been documented to be localized in the Malpighian tubules in other insect species [15]. In *D. melanogaster*, for instance, Val13 is located in the Malpighian tubules. Moreover, for each subunit, exactly one gene is strongly upregulated or its protein is highly abundant in the Malpighian tubules [33].

Secondly, Malpighian tubules in ds*vATPaseG*-fed larvae were clear, in contrast to the opaque tubules from ds*egfp*-fed larvae. An identical Malpighian tubule phenotype has been documented after knockout of V_1_ region subunits A, B, D, E, F, G and H, and V_0_ proteins c, c” and e encoding genes in *D. melanogaster* [33,55]. Correspondingly, human renal disorders have been recorded in the mutants in vATPase genes specifically expressed in kidney [56,57,58].

In insects, the main nitrogenous waste metabolites are excreted as uric acid. Urate ions are transported into the tubule lumen in soluble form [33], where they are precipitated to uric acid crystals below a certain pH value [33,59]. In *H. vigintioctopunctata* (this study) and *D. melanogaster* [59,60], dysfunction of vATPase inhibits sufficient acidification of the lumen in Malpighian tubules. As a result, urate ions cannot become uric acid crystals and the tubule lumen remains translucent.

Thirdly, the HE staining showed that most of the Malpighian tubules in ds*vATPaseG*-fed larvae are empty, whereas the tubules from ds*egfp*-fed larvae were full of urine in *H. vigintioctopunctata*. Given that vATPase-driven accumulation of K^+^ triggers the formation of primary urine [6,14], knockdown of *vATPaseG* inhibits the formation process and leads to the empty Malpighian tubules in *H. vigintioctopunctata*.

## Figures and Tables

**Figure 1 insects-12-00935-f001:**
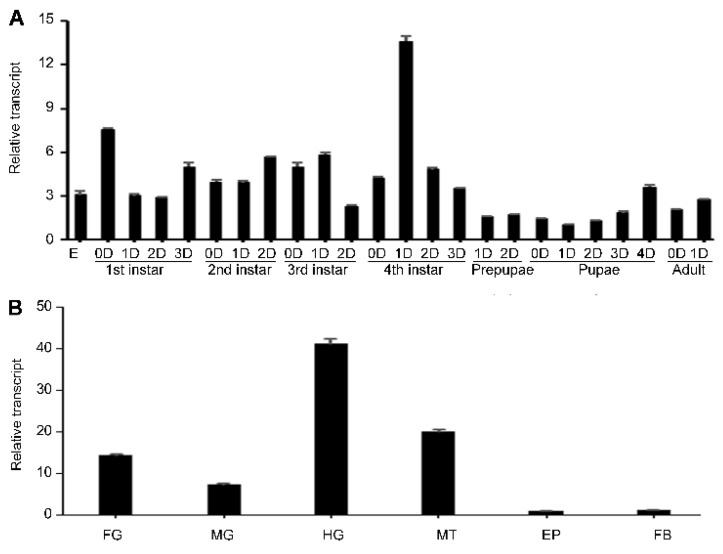
Temporal (**A**) and tissue (**B**) transcription patterns of *HvvATPaseG* in *Henosepilachna vigintioctopunctata*. For temporal expression analysis, RNA templates were derived from eggs (day 3), the larvae from the first through the fourth instars, prepupae, pupae and adults (D0 indicated newly ecdysed larvae or pupae, or newly emerged adults). For tissue expression analysis, the relative transcripts were measured in the foregut (FG), midgut (MG), hindgut (HG), Malpighian tubules (MT), epidermis (EP) and fat body (FB) of the day 1 final instar larvae. For each sample, three independent pools of 20–30 individuals were measured in technical triplicate using qRT-PCR. The values were calculated using the 2^−ΔΔCT^ method. The relative transcripts are the ratios of copy numbers in different developing stages relative to egg, or those in different tissues relative to EP, which are set as 1. The columns represent averages with vertical lines indicating SE.

**Figure 2 insects-12-00935-f002:**
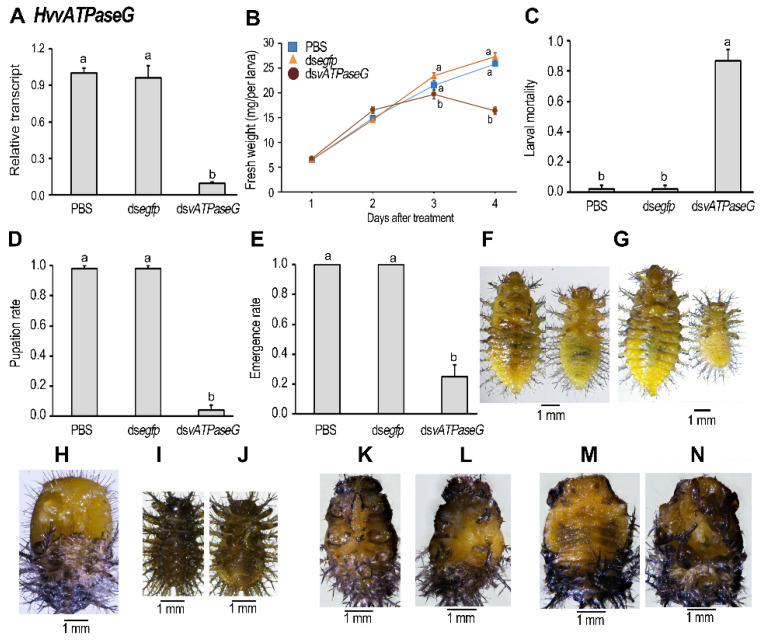
Ingestion of ds*vATPaseG* by the final instar larvae affects performance in *Henosepilachna vigintioctopunctata*. The newly-ecdysed final instar larvae had ingested PBS-, ds*egfp*-, ds*vATPaseG*-dipped leaves for three days. The expression levels of *HvvATPaseG* were determined (**A**). Relative transcripts are the ratios of relative copy numbers in treated individuals to PBS-fed controls, which are set as 1. The fresh weights were measured daily after initiation of the experiment (**B**); the larval sizes were shown 3 and 4 days after initiation of the experiment (**F**,**G**; PBS- vs. ds*vATPaseG*-fed larvae). The larval mortality, pupation and the emergence rate were recorded during a 3 week trial period (**C**–**E**). The bars represent values (±SE). Different letters indicate significant difference at *p* value < 0.05 using analysis of variance with the Tukey-Kramer test. While the CK larvae pupated 6 days after initiation of bioassay (**H**), most RNAi larvae remained as prepupae (**I**–**N**).

**Figure 3 insects-12-00935-f003:**
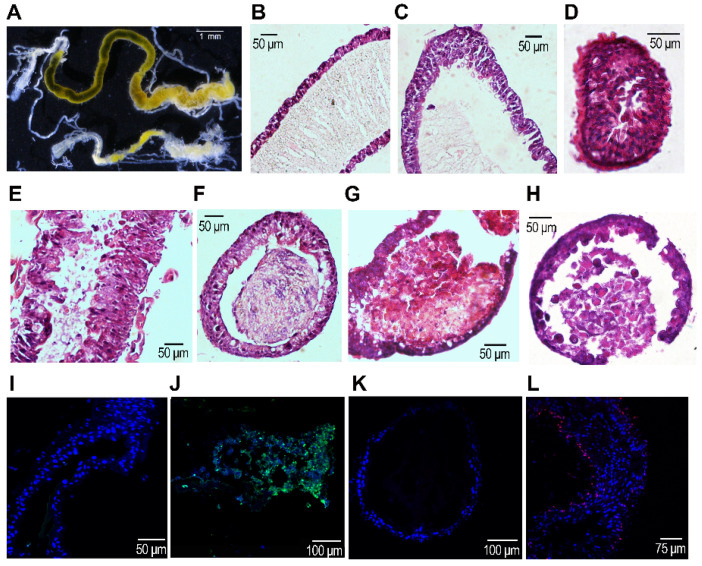
Feeding ds*vATPaseG* affects the structure of midguts in *Henosepilachna vigintioctopunctata*. The newly-ecdysed fourth instar larvae had ingested PBS-, ds*egfp*-, and ds*vATPaseG*-dipped leaves for three days. The guts were dissected 4 days after the initiation of bioassay (**A**). The midgut sections from ds*egfp*- (**B**–**D**) and ds*vATPaseG*-fed (**E**–**H**) larvae were stained with hematoxylin-eosin (HE) method. The midguts from PBS- and ds*egfp*-fed larvae have tall and columnar epithelium cells (EP) and clear peritrophic membrane (PM), in contrast to ds*vATPaseG*-fed larvae. Midgut sections from fourth-instar larvae were stained with DAPI and Hoechst 33342 and to visualize the DNA, and TUNEL kit and 5-ethynyl-2′-deoxyuridine (EdU) Alexa FluorTM 594 to see the apoptotic and actively dividing cells respectively. Merged images of DAPI/TUNEL (**I**,**J**) and Hoechst 33342/EdU staining are shown (**K**,**L**). The TUNEL- and EdU-positive cells in the midgut of ds*vATPaseG*-fed larvae are green and red in color respectively (**J**,**L**), in contrast to DAPI-positive cells (blue) in the midgut of ds*egfp*-fed larvae (**I**,**K**).

**Figure 4 insects-12-00935-f004:**
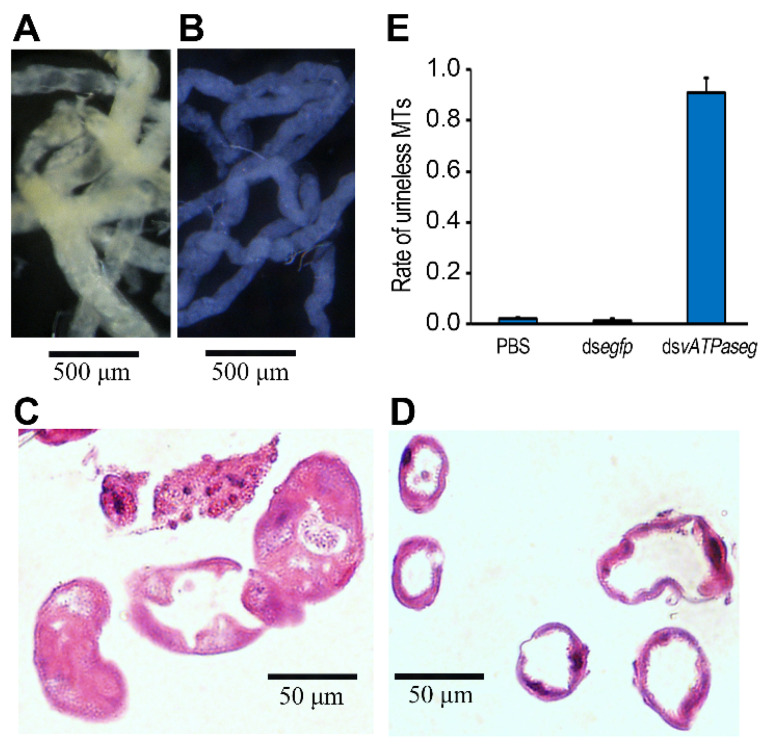
A characteristic phenotype in Malpighian tubules in ds*vATPaseG*-fed larvae in *Henosepilachna vigintioctopunctata*. The newly-ecdysed fourth instar larvae had ingested PBS-, ds*egfp*-, and ds*vATPaseG*-dipped leaves for three days. The Malpighian tubules were dissected and collected 5 days after the initiation of bioassay (**A**). The Malpighian tubules from ds*egfp*-fed larvae are opaque. In contrast, the tubules in ds*vATPaseG*-fed larvae are clear. The hematoxylin-eosin (HE) staining shows that the tubules from ds*egfp*-fed larvae are full of urine (**B**), whereas most of the tubules in ds*vATPaseG*-fed larvae are empty (**C**,**D**). The rates of urineless Malpighian tubules were calculated (**E**).

**Figure 5 insects-12-00935-f005:**
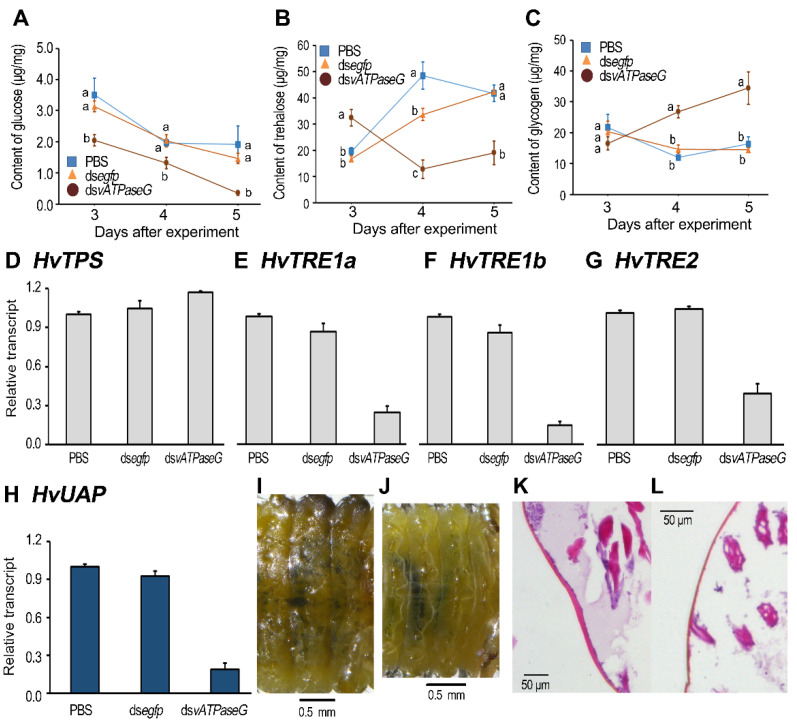
Knockdown of *HvvATPaseG* affects sugar contents in *Henosepilachna vigintioctopunctata*. The newly-ecdysed final instar larvae had ingested PBS-, ds*egfp*-, and ds*vATPaseG*-dipped leaves for three days. The specimens 3, 4 and 5 days after experiment were collected and the contents of glucose, trehalose and glycogen were determined (**A**–**C**). The expression levels of a trehalose biosynthesis gene (*HvTPS*), three trehalose degradation genes (*HvTRE1a*, *HvTRE1b* and *HvTRE2*) and a chitin biosynthesis gene (*HvUAP*) were tested (**D**–**H**). Relative transcripts are the ratios of relative copy numbers in treated individuals to PBS-fed controls, which are set as 1. The bars represent values (±SE). Different letters indicate significant difference at *p* value < 0.05 using analysis of variance with the Tukey-Kramer test. The body cuticle was almost transparent in the ds*vATPaseG*-treated larvae, in contrast to ds*egfp*-ingested ones (**J** vs. **I**). The hematoxylin-eosin (HE) staining shows that the cuticle in the head capsule from PBS- and ds*egfp*-fed larvae are thicker (**K**) than that from ds*vATPaseG*-fed larvae (**L**).

**Figure 6 insects-12-00935-f006:**
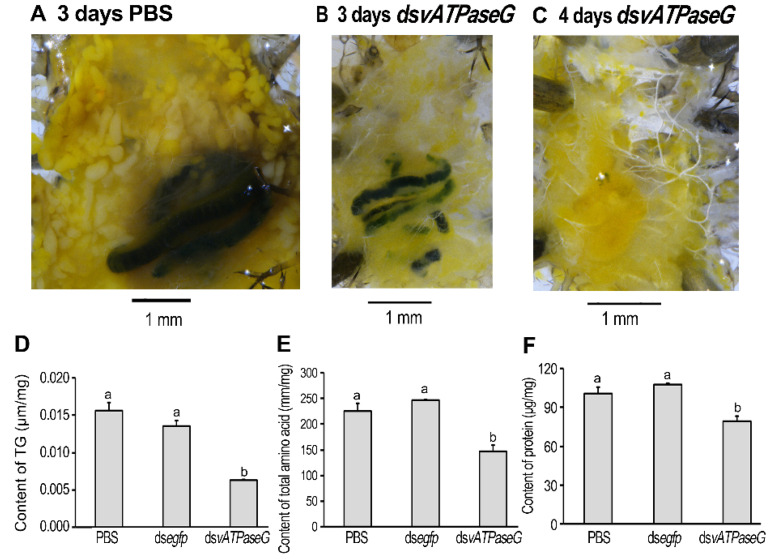
The accumulation of lipid and protein in the *Hv**vATPaseG* RNAi larvae in *Henosepilachna vigintioctopunctata*. The newly-ecdysed fourth instar larvae had ingested PBS-, ds*egfp*-, and ds*vATPaseG*-dipped leaves for three days, and in normal foliage for an additional of 2 days. The larval specimens 3 and 4 days after experiment were collected and dissected (**A**–**C**). The contents of triglyceride, total soluble amino acids and proteins were determined 5 days after experiment (**D**–**F**). The bars represent values (±SE). Different letters indicate significant difference at *p* value < 0.05 using analysis of variance with the Tukey-Kramer test.

## Data Availability

All data generated in association with this study have been made available in the Appendix A published online with this article.

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
