# Peer review of "Knockdown of Vacuolar ATPase Subunit G Gene Affects Larval Survival and Impaired Pupation and Adult Emergence in Henosepilachna vigintioctopunctata"

_insects, 2021, doi:10.3390/insects12100935_

Round 1

Reviewer 1 Report

In this paper, HvvATPaseG gene of Henosepilachna vigintioctopunctata was knocked down by ingestion. Silence of HvvATPaseG affected many physiological index of H. vigintioctopunctata. These results are of great significance for the further functional research of vATPase G subunit in insects. At the same time, it establishes a theoretical foundation for H. vigintioctopunctata control. Following are some problems in this article.

1、The title cannot summarize the research content. Please revise and give an appropriate title.

2、In Figure 3, There are three treatments, PBS, dsegfp and dsvATPaseG, so pictures of these three treatments should all be showed in Figure 3. There are just pictures of dsegfp and dsvATPaseG.

3、L342-343 “the treated larvae after consumption of dsvATPaseG for 3 days were dissected and the Malpighian tubules were collected”, but L351-352 “The Malpighian tubules were dissected and collected 5 days after the initiation of bioassay”. It is confused?

Author Response

Please see the response letter. I have answered all issues there.

Reviewer 2 Report

General comments

In this work, the authors demonstrated that the knock-down of vATPase G subunit gene expression of the coleoptera Henosepilachna vigintioctopunctata arrested its development, inhibited pupation, and impaired the adult emergence.  Therefore, they suggested that the silencing of vATPase G subunit (by using RNAi approach) in this insect can be an alternative for its control. The work is relevant because the insect H. vigintioctopunctata is a serious defoliator of Solanaceae and Cucurbitaceae plants worldwide. Furthermore, to the best of my knowledge, functional characterization of vATPase subunit G has been documented only for D. melanogaster. The article is well written, the methodology is carefully described and the cited literature is up to dated. All experiments were properly controlled and the results support the conclusions.

Specific comments

In the Discussion, lines 468-69, the authors mentioned that “It is known that high pH value is essential for maintenance of high enzymatic activities in the midguts”, and cited the reference [19]. However, in that article the author mentioned caterpillars and mosquitoes, not coleoptera. It is well know that coleoptera have mainly cysteine peptidases in their midguts, and these enzymes work on pH <6. Could the authors clarify this part?

Item 2.3. (Preparation of DsRNAs): I would like to have seen a figure (even that as supplementary), from the agarose gel showing the dsRNA. Could the authors provide it? By the way, how did the authors determine the dsRNA concentration of about 0.5 μg/ml?

Minor

Line 112: Please correct “TBLASTIN” (to TBLASTN)

Line 167: Please cite the DNAse I source

Line 408: Change “update” by “Up to date”

In my opinion, the citation of figures between parentheses in the Discussion (e.g. Figure 3 was cited many times) is not necessary, since they were cited in the Results.

Line 488: Consider using “research” instead of “survey”

Author Response

Please see the response letters. I have answered all the issues here.
